# Resveratrol, EGCG and Vitamins Modulate Activated T Lymphocytes

**DOI:** 10.3390/molecules26185600

**Published:** 2021-09-15

**Authors:** Joseph Schwager, Nicole Seifert, Albine Bompard, Daniel Raederstorff, Igor Bendik

**Affiliations:** Department of Human Nutrition & Health, DSM Nutritional Products Ltd., P.O. Box 2676, CH-4002 Basel, Switzerland; nicole.seifert@dsm.com (N.S.); albine.bompard@dsm.com (A.B.); raederstorff.daniel@orange.fr (D.R.); igor.bendik@dsm.com (I.B.)

**Keywords:** resveratrol, EGCG, vitamin A, vitamin D, vitamin E, adaptive immune response, T lymphocytes, in vitro activation, cytokines, interleukins, synergism/antagonism

## Abstract

Vitamins and bioactives, which are constituents of the food chain, modulate T lymphocyte proliferation and differentiation, antibody production, and prevent inflammation and autoimmunity. We investigated the effects of vitamins (vitamin A (VA), D (VD), E (VE)) and bioactives (i.e., resveratrol (Res), epigallocatechin-3-gallate (EGCG)) on the adaptive immune response, as well as their synergistic or antagonistic interactions. Freshly isolated T lymphocytes from healthy individuals were activated with anti-CD3/CD28 antibodies for 4–5 days in the presence of bioactives and were analyzed by cytofluorometry. Interleukins, cytokines, and chemokines were measured by multiple ELISA. Gene expression was measured by quantitative RT-PCR. Res and EGCG increased CD4 surface intensity. EGCG led to an increased proportion of CD8^+^ lymphocytes. Anti-CD3/CD28 activation induced exuberant secretion of interleukins and cytokines by T lymphocyte subsets. VD strongly enhanced T_h_2 cytokines (e.g., IL-5, IL-13), whereas Res and EGCG favored secretion of T_h_1 cytokines (e.g., IL-2, INF-γ). Res and VD mutually influenced cytokine production, but VD dominated the cytokine secretion pattern. The substances changed gene expression of interleukins and cytokines in a similar way as they did secretion. Collectively, VD strongly modulated cytokine and interleukin production and favored T_h_2 functions. Resveratrol and EGCG promoted the T_h_1 response. VA and VE had only a marginal effect, but they altered both T_h_1 and T_h_2 response. In vivo, bioactives might therefore interact with vitamins and support the outcome and extent of the adaptive immune response.

## 1. Introduction

The interactions between cell populations such as T and B lymphocytes orchestrate the highly complex pattern of the immune response to pathogens. T lymphocytes differentiate into two main distinct subsets of T effector cells, T helper type 1 (T_h_1) and T helper type 2 (T_h_2) cells (reviewed in: [1]). T_h_1 cells secrete IL-2, IFN-γ, and TNF-α and are important for the development of delayed type hypersensitivity reactions and protective responses to intracellular pathogens [2]. T_h_2 lymphocytes express and secrete IL-4, IL-5, and/or IL-13 and are essential for the development of humoral and allergic reactions. The cytokine milieu of the local microenvironment is a major determinant of the direction of T_h_ cell differentiation. Cytokines, including IL-12 and IFN-γ, directly induce progenitor (p) T_h_ cell differentiation into T_h_1 cells, whereas IL-4 stimulates pT_h_ cell differentiation into T_h_2 cells. Recent evidence also suggests an important role for cytokines such as IL-1α, IL-1β, IL-15, and IL-18 in stimulating T_h_1 responses [2] and IL-10 and IL-13 in stimulating T_h_2 responses [3].

Vitamins and bioactives have long been known to modulate adaptive immune reactions [4,5]. Vitamin D (VD) and, in particular, the VD metabolite 1,25(OH)_2_VD_3_ has potent immune-regulatory effects and, thus, an important role in maintaining immune homeostasis. VD inhibits CD4^+^ T_h_1 proliferation, the expression of IL-2 and INF-γ and CD8^+^ T cell-mediated cytotoxicity. VD exerts a potent action on T_reg_ cells and their secreted cytokines and interleukins. VD mitigates the production of T_h_1 signature cytokines [6] and promotes the secretion of T_h_2 cytokines, but it also regulates the innate immune cells, since it stimulates human monocytes proliferation and differentiation [7]. Vitamin A (VA) and metabolically produced VA-retinoids are potent modifiers of rodent T_h_1 and T_h_2 responses [8,9]. Several mechanisms were proposed to account for these observations, including the direct downregulation of T cell IFN-γ synthesis, direct promotion of T_h_2-cell differentiation, and/or alteration of accessory or antigen presenting cell function toward a T_h_2-inducing phenotype [10]. Vitamin E (or a-tocopherol) (VE) is a potent antioxidant vitamin that diminishes the release of pro-inflammatory cytokines and chemokines and modulates cellular immune function and cell adhesion. It reduces the production of reactive oxygen species (ROS), most likely via the NF-κB activation pathway [11,12,13]. Several studies have revealed that green tea extracts containing EGCG modulate T lymphocyte activity [14,15,16,17,18]. Similarly, Res shapes CD4^+^ and CD8^+^ lymphocyte activity and has dose-dependent stimulatory or inhibitory activities on the T lymphocyte immune response [19,20,21,22,23,24,25,26,27].

In this study, we evaluated the relative contribution of vitamins and bioactives to interleukin and cytokine production in activated human T lymphocytes, and we investigated their effects on activated human T lymphocyte function.

## 2. Results

### 2.1. Phenotype of Activated PBMCs

T lymphocytes and subpopulations thereof (i.e., CD3^+^, CD4^+^, and CD8^+^ T lymphocytes) were isolated from PBMCs by negative selection and then analyzed by cytofluorometric analysis. Isolated PBMCs contained 27% ± 3% CD^8+^ and 59% ± 2% CD4^+^ lymphocytes (*n* = 3). Selected CD3^+^ lymphocytes were T lymphocytes, since about 60% and 35% of these cells were CD4^+^ and CD8^+^, respectively. CD3^+^ cells were CD23^+^/CD18^−^, CD86^−^, CD11c^−^, and CD14^−^, which reflects the absence of B cells and monocytes/macrophages. Furthermore, ~40% of CD3^+^ cells expressed the αβ TCR (results not shown). Negatively selected CD4^+^ or CD8^+^ lymphocytes were >95% CD4^+^ and ~80–90% CD8^+^, respectively (Appendix A). CD4^+^ lymphocytes were CD18^+^ and CD40^−^, which indicated the absence of B lymphocytes. A substantial proportion of freshly isolated CD4^+^ lymphocytes expressed CCR4 (not shown); CD8^+^ selected lymphocytes were CD18^+^ and CD4^−^.

### 2.2. In Vitro Differentiation of Activated T Lymphocytes

Blood cells were cultured with anti-CD3/CD28 (immobilized on Dynabeads^™^) to induce T lymphocyte proliferation and differentiation, see [28]. Cytofluorometric analysis was performed after 5 days of culture. The proportion of CD4^+^ and CD8^+^ lymphocytes was similar in both unstimulated and activated cells. Anti-CD3/CD28 stimulation induced the expansion of the CCR4^+^ cell population (Figure 1). The activated cells were TCRαβ^+^, but they did not express TCRγδ and TLR4 determinants (not shown). The data indicated that the entire CD3^+^ lymphocyte population expanded after anti-CD3/CD28 activation. We also prepared CD4^+^ and CD8^+^ lymphocytes and found that both cell populations vigorously proliferated when stimulated with anti-CD3/CD28. Both T lymphocyte subsets conserved their respective phenotypes during the entire culture period. Anti-CD3/CD28 stimulated cells expressed increased cell surface density of CD4 and CD8 determinants (compared to non-activated cells).

### 2.3. Effects of Res, EGCG, and Vitamins on the Phenotype of In Vitro Activated T Cells

We investigated whether anti-CD3/CD28 activated T lymphocytes had an altered phenotype when they were cultured in the presence of substances. Res or EGCG did not markedly alter CD4 surface expression, but it increased expression when both substances were combined (Figure 1 and Appendix A). VD had an opposing effect and reduced mean surface intensity of CD4 on activated T lymphocytes. VA and VE had no significant effects on CD4 or CD8 surface intensity or percentage of positive cells (results not shown). EGCG slightly shifted the CD4/CD8 ratio to an increased proportion of CD8^+^ cells (Appendix A). In contrast, Res, VA, VD, and VE had no impact on the CD4/CD8 ratio in stimulated T cells (Appendix A). The combination of VD with Res or EGCG further altered CD8 surface expression; Res and EGCG, alone or combined, favored a high level of CCR4 expression, a marker for T_h_1 lymphocytes. Conversely, VD reduced its surface density (Figure 1).

### 2.4. Cytokines Produced by In Vitro Activated T Lymphocytes

Activated T lymphocytes differentiated into CD4^+^ T_h_ cell subsets, each of which produced a genuine set of T_h_ lineage signature cytokines and chemokines [2]. Similarly, CD8^+^ lymphocytes preferably secreted cytokines, which are instrumental for cellular immune functions. We investigated the changes of secreted cytokines by activated T lymphocytes and CD4^+^ or CD8^+^ lymphocyte subsets. Anti-CD3/CD28 activation induced exuberant secretion of interleukins and cytokines, which mirror the in vitro differentiation of T_h_ lymphocyte subsets. INF-γ and IL-2 were prototypic for the T_h_1 compartment, whereas IL-5 and IL-13 were distinctive for activated T_h_2 lymphocytes (Appendix A). Activated T lymphocytes also produced large amounts of chemokines, including CCL5/RANTES, CXCL8/IL-8, MIP-1a/CCL3, and MIP-1β/CCL4 (Appendix A). Similarly, isolated CD4^+^ or CD8^+^ lymphocytes secreted substantial amounts of cytokines upon activation with anti-CD3/CD28. Compared to CD8^+^ lymphocytes, activated CD4^+^ cells produced significantly more IL-2, IL-6, IL-9, IL-10, and IL-17, and TNF-αcells produce^+^ lymphocytes out-performed CD4^+^ cells in the production of IL-5, IL-13, and various chemokines (Appendix A).

### 2.5. Selective Effects of Vitamins and Polyphenols on Cytokines and Interleukins Produced by Activated T Lymphocytes

The presence of substances during lymphocyte activation and differentiation influenced interleukin and cytokine production. Res significantly increased the production of IL-2 (Figure 2A). It also augmented the production of IL-6, whereas it blunted chemokine CXC/CL8 production (Figure 2D,K). VD drastically enhanced IL-13 secretion of activated T lymphocytes and significantly augmented IL-5 and IL-6 production (Figure 2E,G,K). VD, however, was less active than its physiological metabolite 1,25(OH)_2_D_3_ (Appendix A). VA promoted the production of IL-2 and IL-5 (Figure 2A,E,I). Retinoic acid (used at 0.01–1 nM) had similar effects on IL-2 (Appendix A). VE altered cytokines and chemokines produced by activated T lymphocytes at high concentration (i.e., 25 μM). Since changes induced by one substance might be counter-balanced or enforced by concomitant changes of another substance, we determined how the substances influenced the ratio of secreted cytokines and, therefore, the T_h_1/T_h_2 balance. Resveratrol significantly increased the IL-2/IL-13 ratio, which is characteristic of an increased T_h_1 response (Table 1). Regarding T_h_1 cytokines, Res had a higher impact on IL-2 compared to INF-γ. Similarly, Res also increased IL-2 production relative to chemokines (CCL5/RANTES, CXCL10/IP-10) and IL-6 (Table 1). EGCG and Res had many similar effects on chemokines. VA and VE had only minor effects on these ratios. VD induced drastic changes in the ratio of prototype T_h_1 and T_h_2 interleukins since it significantly up regulated the T_h_2 at the expense of decreased T_h_1 interleukin production (Table 2). This is also reflected in the ratio for cumulative prototype T_h_1 and T_h_2 cytokines, which were defined as IL-2, IFN-γ and IL-5, IL-13, respectively.

Both CD4^+^ and CD8^+^ lymphocytes responded to anti-CD3/CD28 stimulation. Yet, the extent of activation was different, since activated CD4^+^ lymphocytes secreted higher amounts of cytokines and interleukins than CD8^+^ lymphocytes did (Appendix A). Vitamins and polyphenols had similar effects on activated CD4^+^ and CD8^+^ lymphocyte subsets (Appendix A).

### 2.6. Interactions between Vitamins and Polyphenols on Cytokine Production by Activated T Lymphocytes

Res and VD had the strongest, and often opposed, effects on cytokine secretion. We investigated whether one of these bioactives had a predominant influence. To this aim, the effect of combinations of Res and VD was evaluated on the secretion pattern of activated T lymphocytes. T_h_1-specific cytokine production was dominated by VD rather than by Res (Figure 3). For instance, VD reduced the enhancing effects of Res on IL-2 production to VD-specific levels. Alternatively, INF-γ production was similar for VD-only conditions and in combinations of VD and Res. This reflects a prevailing effect of VD. The level of T_h_2-specific IL-13 was dominated by VD. We observed synergistic effects between Res and VD on IL-13 production, since it exceeded the sum of single effects of Res and VD (Figure 3). Regarding TNF-α or CXCL8/IL-8, the combinations of the two substances generated an intermediate pattern of cytokine production. Other combinations of substances (e.g., VD with VE, VD with VA, VD with EGCG) corroborated the dominant effect of VD over other vitamins and bioactives (data not shown).

### 2.7. Effects of Vitamins and Polyphenols on Gene Expression of Activated T Lymphocytes

Some of the tested substances considerably influenced cytokine gene expression (Figure 4). The most prominent effects were observed with VD, since it blunted expression levels of IL-2, INF-γ, but also IL-17, IL-21, and TNF-α. In contrast, it significantly increased gene expression of T_h_2 prototype interleukins, such as of IL-5 and IL-13, whereas IL-10 expression was moderately changed. IL-6, which promotes T_h_2 and B cell differentiation [29], was strongly enhanced by VD. Res, EGCG, VE, and VA had weak effects on cytokine gene expression. However, Res and EGCG increased T_h_1 specific IL-2 and INF-γ.

## 3. Discussion

This study shows that bioactives and vitamins modulate the composition of T lymphocyte subsets and significantly influence the secretion of interleukins and cytokines. Activation of peripheral blood leukocytes, or subsets thereof, with anti-CD3/CD28 induced the expansion of T lymphocytes. The presence of EGCG slightly favored the proliferation of CD8^+^ lymphocytes. The impact of VA, VD, VE, and Res on the CD4^+^/CD8^+^ ratio was minor. Yet, the combination of VD and EGCG, or Res modulated surface density of CD4 and CD8 determinants and it promoted the expression of CCR4. This receptor is a marker of T_h_2 subsets; thus, the observed phenotypic changes are consistent with an increased T_h_2 response, which might be mainly orchestrated by VD [32]. These changes suggest that bioactives and vitamins, as well as combinations thereof, alter the T lymphocyte compartment. They may influence the overall CD4/CD8 balance as well as the differentiation of T lymphocyte subsets. It should be noted, that among vitamins, VD rather than VA and VE showed these effects on T lymphocyte compartments.

The underlying cellular mechanisms that trigger these changes are numerous and presumably dependent on the cell types involved. Res and EGCG impaired the production of reactive oxidant species (ROS) (reviewed by [33,34]); VE also influenced ROS production [35,36]. As a consequence, cytokines and inflammatory interleukins like TNF-α and IL-1α/β are mitigated by the presence of Res or EGCG [30,37,38] in a cell- and tissue-dependent manner [39]. It should be noted that Res up regulated IL-6 in most of these compartments [39]. Conversely, Res reduced T cell activation in murine spleen cells [40]. The generation of ROS may be implicated in the favorable growth of T_reg_ cells in the presence of resveratrol [41]. As described previously [38,42] Res differently modulated macrophages, spleen cells, and peripheral blood leukocytes. In line with this observation, Res had pleiotropic effects on gene and protein expression and reduced pro-inflammatory cytokine release from T-lymphocytes [21], and it altered nuclear factors essential to the process of lymphocyte differentiation [43].

EGCG was also shown to influence cell growth in lymphoid cells and eventually the immune system ([31,44], reviewed in [45]). We observed only minor effects of VA and its physiological correlate RA on the adaptive immune response. As shown in previous studies, RA activated effector CD4^+^ T lymphocytes via the production of IFN-γ by lymphocytes of the innate lymphoid cells [46]. It also influenced T_h_2 responses in human CD4^+^ Tlymphocytes in vitro and in vivo [9,41,47].

VE, which has anti-inflammatory and antioxidant properties [35], showed favorable effects on T lymphocytes during immunosenescence; in vivo studies revealed that dietary supplementation with VE improved the immune response of T lymphocytes in aged animals or individuals [11,12,13]. It also influences immune function through modulating cAMP levels, and ultimately PGE_2_ [36]. The moderate effect of VE in this study might be due to a relatively poor cellular ‘loading’ during in vitro T lymphocyte activation.

As shown in numerous seminal studies, VD has a variety of effects on the adaptive immune response [48,49,50,51,52,53,54,55,56], as well as on the innate immune response. Historically, VD was first associated with immunosuppression, while more recent evidence demonstrated its impact on T regulatory (T_reg_) lymphocytes. There is a possible pathway for crosstalk between VD and Res at the VDR signaling [57]. Interactions between Res and VD have also been elucidated in the innate immune response [58].

In this study we have evidenced the interactions between bioactives and vitamins. While the effect on the T_h_1/T_h_2 response revealed some antagonism between Res and VD, we also observed synergistic interactions between the two substances, e.g., in the expression of IL-13; other interactions were additive (Figure 3). This is in line with published data, which showed interactions between bioactives and/or vitamins [44,59,60,61,62].

The present study indicates that Res and VD modulate the activity of T_h_1 and T_h_2 subsets, respectively. The two substances act on distinct and complementary arms of the immune response, since Res increased T_h_1 immunity, while VD favored T_h_2 responses. The two substances profoundly differ in their bioavailability and plasma kinetics. Res, as well as EGCG, are metabolized within hours after dietary intake [63,64,65], whereas VD levels remain high for prolonged periods after uptake [66]. Cellular levels of Res and EGCG might exceed plasma levels (see also [67]). We anticipate that these distinct kinetics offer an approach for targeting the effects of polyphenols versus those of VD on the immune response; due to very short half-life in plasma, Res or EGCG can only briefly influence the immune response, while VD has long-lasting effects. Res and VD react with different cell-based molecules or receptors, which might enable dichotomic cellular effects. The biological consequence of opposite effects of Res and VD, as well as their specific temporal pattern, might result in fine-tuning the adaptive immune response. This needs to be corroborated in future studies.

## 4. Methods

### 4.1. Reagents

Ethanol and DMSO were from Sigma (Sigma-Aldrich, Buchs, Switzerland). Vitamin A (VA), vitamin E (i.e., all-rac α-tocopherol) (VE), retinoic acid (RA), 1,25(OH) vitamin D_3_, and epigallocatechin 3-gallate (EGCG) were from Sigma (Sigma-Aldrich, Buchs, Switzerland); 25(OH) vitamin D_3_ (VD) was from DSM Nutritional Products (Kaiseraugst, Switzerland); trans-resveratrol (Res) was from Sigma (Sigma-Aldrich, Buchs, Switzerland) or from DSM Nutritional Products (Kaiseraugst, Switzerland). All substances were >99% pure. Compounds were dissolved in ethanol (VA, RA, VD, VE) or in DMSO (Res, EGCG) and added to the culture medium concomitantly with the stimulus. Final ethanol and DMSO concentration were 0.05% and 0.5%, respectively.

Lymphoprep^®^ was from Axis Shield AS (Oslo, Norway). Depletion Dynabeads^®^ for isolating human T cells, CD4^+^, or CD8^+^ lymphocytes by negative selection were from Life Technologies Europe B.V (Zug, Switzerland). Human recombinant interferon-γ (IFN-γ) was from Preprotech EC (London, UK). Primers and probes used in RT-PCR were designed with the Primer Express^TM^ program (Applied Biosystems Inc., Foster City, CA, USA) and synthesized by Sigma (Sigma-Aldrich, Buchs, Switzerland). Dynabead immobilized anti-CD3/anti-CD28 (i-antiCD3/CD28) were from Invitrogen (Life Technologies Europe B.V. Zug, Switzerland).

Fluorochrome-conjugated monoclonal antibodies were purchased from BD Pharmingen (San Diego, CA, USA) or eBioscience (Vienna, Austria) and used according to the manufacturers’ indications. Monoclonal antibodies against CD4, CD11c, CD23, CD86, CCR3, CCR6, TCRα/β, and TLR2 were conjugated with FITC; monoclonal antibodies against CD8, CD14, CD18, CCR4, CCR5, TCRγ/δ, and TLR4 were conjugated with PE.

### 4.2. Peripheral Blood Leukocytes and Isolation of Human T Lymphocytes

Buffy coats were prepared from blood obtained from healthy humans at the Blood Donor Center University Hospital (Basel, Switzerland). Peripheral blood mononuclear cells (PBMCs) were isolated by Lymphoprep^®^ within <3 h after blood withdrawal and used for experiments immediately. T lymphocytes were further isolated from PBMCs using the Dynabeads^TM^ Untouched^TM^ Human T cells kits. T lymphocytes, CD4^+^, and CD8^+^ lymphocytes were isolated by negative selection using Depletion Dynabeads^®^ Untouched^TM^ Human T cells, Untouched^TM^ Human CD4 T cells, and Untouched^TM^ Human CD8 T cells (Life Technologies Europe B.V., Zug, Switzerland), respectively, following the experimental protocols provided by the manufacturer. In some cases, isolated PBMC were cryopreserved in fetal bovine serum (FBS) containing 10% DMSO (both from Sigma-Aldrich, Buchs, Switzerland) and stored in liquid nitrogen before being used for experiments.

### 4.3. Activation of T Lymphocytes with Immobilized Anti-CD3/CD28

Cells were cultured in OpTmizer^TM^ CTS^TM^ T-Cell Expansion SFM. Cell viability was determined by the Trypan Blue exclusion test and exceeded 95%. For in vitro cultures, cells were adjusted to 1 × 10^6^ cells/mL. Human T lymphocytes, CD4^+^ and CD8^+^ lymphocytes (5 × 10^5^ cells per culture) were activated with Dynabeads Human T-Activator CD3/CD28 (2.5 μL beads per 10^5^ cells). All reagents were from Life Technologies Europe B.V., Zug, Switzerland. Cells were cultured for 5 days and analyzed by FACS. Culture supernatants were analyzed by multiplex ELISA as described previously [67]. Substances (vitamins, polyphenols) were added to cultures concomitantly to stimulation with immobilized anti-CD3/CD28.

### 4.4. Cell Cytofluorometry

Cells were resuspended in HBSS/2% FCS/0.01% NaN_3_ (HFN) and incubated with fluorochrome-conjugated antibodies for 45 min at 4 °C. Subsequently, cells were washed 3× in HFN and resuspended in HFN containing 7-amino-actinomycin D (7-AAD) (2.5 μg/mL). Data of live (i.e., 7-AAD) cells were acquired with a FACS Calibur cytofluorometer (Becton Dickinson, Allschwil, Switzerland) and evaluated with Cellquest software (Becton Dickinson, Mountain View, CA, USA) as described [67].

### 4.5. Measurement of Secreted Cytokines and Metabolites

Secreted proteins and metabolites were quantified using multiplex ELISA kits obtained from Bio-Rad Laboratories (Hercules, CA, USA) and used in the LiquiChip Workstation IS 200 (Qiagen, Hilden, Germany) as described [39]. Data evaluation was done using the LiquiChip Analyser software (Qiagen, Hilden, Germany).

### 4.6. Statistical Analysis

Data are presented as mean ± SD or ± SEM. The difference between means was assessed by the student test and ANOVA using the SPSS software (IBM SPSS v22, Dynelytics AG, Zürich, Switzerland); *p* values of <0.05 were considered to reflect statistically significant differences [39].

### 4.7. Measurement of Gene Expression

Total RNA was isolated from cells cultured for 4 days (Reference added in proof: T. Sekiya & A. Yoshimura. In Vitro Th Differentiation Protocol. In: Xin-Hua Feng et al (eds.), TGF-β Signaling Methods and Protocols Methods in Molecular Biology, vol 1344, DOI 10.10007/978-1-4939-2966-2, Springer Science-Business Media New York, 2016) and reverse-transcribed as detailed before [38,68]. Real-time PCR analysis was performed using the ABI 7900HT Fast Real-Time PCR System (ThermoFisher, Foster City, CA, USA). The 18S rRNA primers and probes were internal standards. Relative gene expression was quantified by subtracting threshold cycles (C_T_) for ribosomal RNA from the C_T_ of the targeted gene (∆C_T_). Relative mRNA levels were then calculated as 2^∆∆CT^, where ∆∆^CT^ refers to the ∆^CT^ of unstimulated minus ∆^CT^ treated cells [38]. Customized low density micro-arrays (LDA) were from Applied Biosystem ABI (Thermo Fisher Scientific, Waltham, MA, USA).

## 5. Conclusions

Vitamins, in particular vitamin D and bioactives like resveratrol and EGCG, alter the phenotype and function of T lymphocytes. Bioactives distinctly enforce or counterbalance the immune-regulatory effect of vitamin D. Since bioactives and vitamin D substantially differ in their physiological half-life, these substances might have distinct und mutually exclusive effects on the adaptive immune responses.

## Figures and Tables

**Figure 1 molecules-26-05600-f001:**
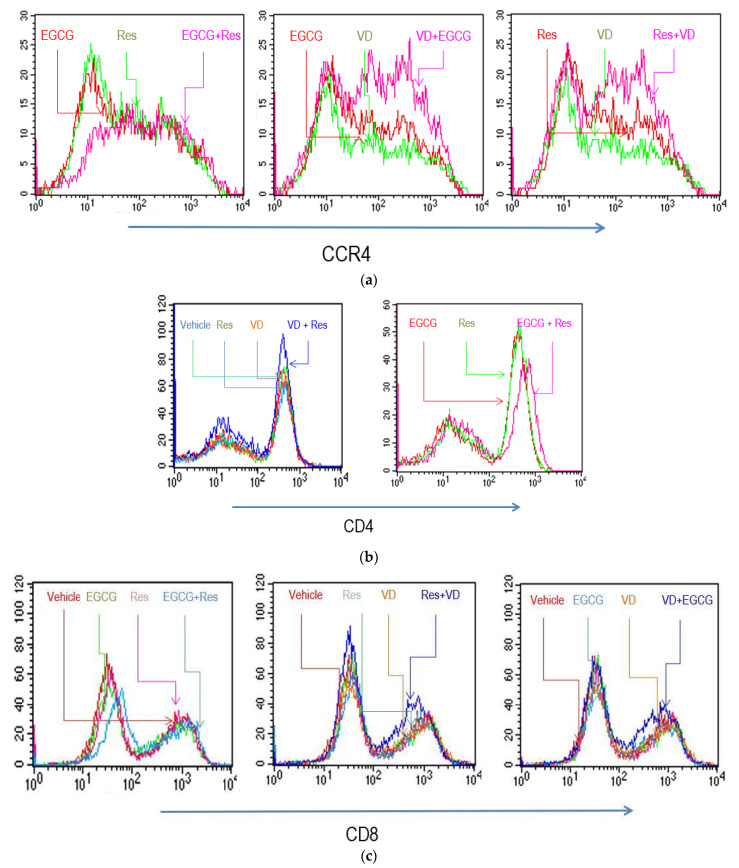
Influence of substances on lymphocyte surface determinants. Anti-CD3/CD28 activated PBMCs were cultured for 5 days, see [20], in the presence of indicated substances and the cytofluorometric profiles were determined. Where not indicated, similar staining pattern was observed in the absence of substances as with EGCG or Res. Cytofluorometric profiles were obtained by incubating cells with anti-CCR4 (**a**), anti-CD4 (**b**) and anti-CD8 (**c**).

**Figure 2 molecules-26-05600-f002:**
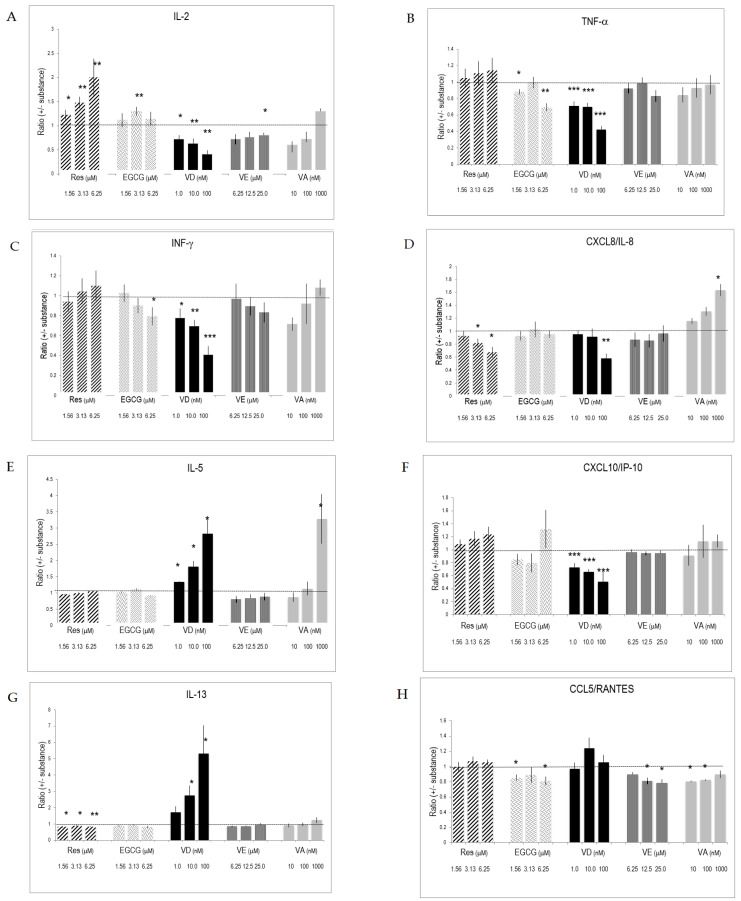
Cytokines and interleukins produced by activated T lymphocytes and the impact of substances on their production. CD3^+^ cells were stimulated for 5 days with anti-CD3/CD28 in the presence of various concentrations of substances. Cytokines, interleukins, and chemokines secreted by triplicate cultures were determined by multiplex ELISA. The data were normalized against values obtained from vehicle-treated cultures. Bars indicate SD (*n* = 3). * = *p* < 0.05; ** = *p* < 0.01; *** = *p* < 0.001. (**A**–**K**) refers to measured secretion of IL-2, TNF-α, INF-**γ**, CXCL8/IL-8, IL-5, CXCL10/IP-10, IL-13, CCL5/RANTES, IL-10, CCL4/MIP-1 and IL-6, respectively. The dotted line in the graphs indicates the level of the reference ratio (=1).

**Figure 3 molecules-26-05600-f003:**
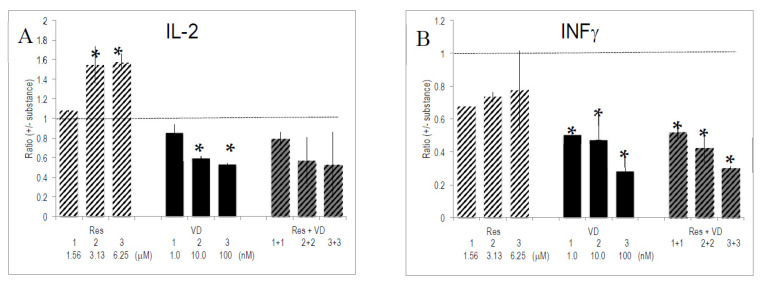
Effect of combinations of substances on cytokine/interleukin production. Anti-CD3/CD28 activated PBMCs were cultured for 4 days (see: reference added in proof) with the indicated substances and combinations thereof. Gene expression was quantified by PCR and the ratio of expression (+/− substances) was computed as described in Materials and Methods. Bars indicate SEM (*n* = 4 experiments with donors of PBMCs, each done in triplicate cultures). * indicate *p* values <0.05 versus anti-CD3/CD28 stimulated PBMCs. (**A**–**E**) refers to the ratio of secreted IL-2, INF-γ, IL-13, TNF-α and CXCL8/IL-8, respectively.

**Figure 4 molecules-26-05600-f004:**
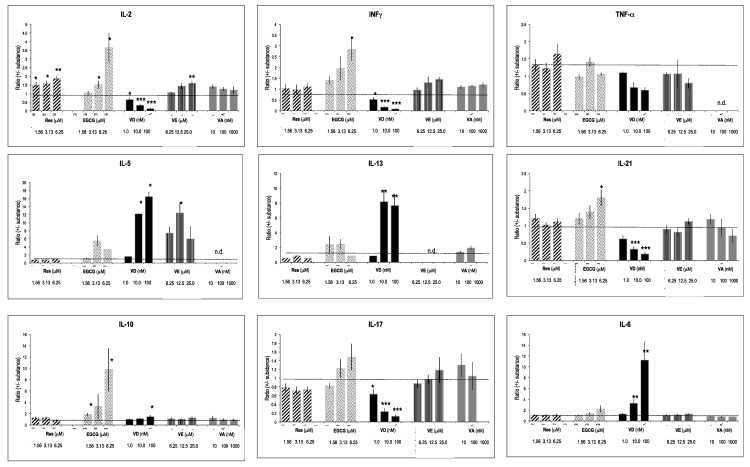
Effect of substances on gene expression. PBMCs were cultured for 4 days (see: reference added in proof), and the gene expression was quantified by real-time PCR. Fold-changes (versus un-activated cells) were determined (see [30,31]). The effect of substances was computed as a ratio (fold-changes in the presence of substances/fold-change with vehicle only). Bars indicate SD (*n* = 4). * = *p* < 0.05; ** = *p* < 0.01; *** = *p* < 0.001. The dotted line in the graphs indicates the level of the reference ratio (=1).

**Table 1 molecules-26-05600-t001:** Effect of substances on the T_h_1/T_h_2 ratio.

	Cumulative Prototype T_h_1/T_h_2 ^(1)^	IL-2/IL-13	IL-2/INF-γ	INF-/IL-5	IL-2/TNF-γ	IL-2/CCL5	IL-2/IL-6
+Res	1.39 ± 0.19 ^(2)^	2.45 ± 0.19	1.47 ± 0.25	2.08 ± 0.61	2.10 ± 0.21	1.78 ± 0.21	2.41 ± 0.21
+EGCG	1.52 ± 0.24	1.37 ± 0.24	0.78 ± 0.39	2.52 ± 0.82	1.74 ± 0.07	1.19 ± 0.41	1.62 ± 0.07
+VD	0.15 ± 0.08	0.13 ± 0.08	6.31 ± 4.85	0.20 ± 0.08	0.90 ± 0.61	0.94 ± 0.53	1.09 ± 0.61
+VA	1.15 ± 0.15	1.40 ± 0.15	1.99 ± 0.47	0.85 ± 0.20	2.91 ± 0.21	1.95 ± 0.34	2.86 ± 0.21
+VE	0.42 ± 0.02	0.52 ± 0.02	1.45 ± 0.18	0.32 ± 0.10	0.73 ± 0.14	1.22 ± 0.03	nd ^(3)^

Secreted interleukins and cytokines of PBMCs, which were activated with anti-CD3/CD28 and cultured for 5 days in the presence or absence of indicated substances, were measured (in triplicates). Data were normalized against ‘activated cells’ (which were set as 1). Mean values ± SEM are given (*n* = 4). ^(1)^ (prototype T_h_1: IL-2, INF-γ)/prototype T_h_2: IL-5, IL-13). ^(2)^ ratio of secreted interleukin or cytokine $=\frac{\left(\mathrm{Activated}\;\mathrm{cells}+\mathrm{substance}\right)}{\mathrm{Activated}\;\mathrm{cells}\;\mathrm{only}}$. ^(3)^ not done.

**Table 2 molecules-26-05600-t002:** Substances induced shift in the T_h_1 and T_h_2 prototype interleukin secretion.

Cells Stimulated in the Presence of	Ratio IL-2/IL-13 (± SEM) ^(1)^		Ratio Cumulative T_h_1/T_h_2 (± SEM) ^(^^2)^	
no substance	1 (ref)		1 (ref)	
Res	2.45 ± 0.19	*	1.39 ± 0.19	*
EGCG	1.37 ± 0.24		1.52 ± 0.24	*
VD	0.13 ± 0.08	*	0.15 ± 0.08	*
VA	1.40 ± 0.15		1.15 ± 0.15	
VE	0.52 ± 0.02		0.42 ± 0.02	

PBMCs were activated with anti-CD3/CD28 and cultured for 5 days. Interleukins were quantified in culture supernatants, normalized (against ‘no substance’), and the ratio was calculated as indicated. ^(1)^ Ratio of secreted interleukins = activated cells + substance/activated cells only. ^(^^2)^ Th1: IL-2 (+) IFNγ; Th2: Il-5 (+) IL-13. * *p* < 0.05 (vs. ref).

## Data Availability

The data presented in this study are available in *Molecules*.

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
