# Peer review of "Resveratrol, EGCG and Vitamins Modulate Activated T Lymphocytes"

_molecules, 2021, doi:10.3390/molecules26185600_

Round 1

Reviewer 1 Report

The impact of oral nutrients and bioactives on immune responses is an important focus deserving of much coverage in the research literature.  This paper contributes to this topic.  A few items that should be addressed by the authors are listed below. 

The use of the terms ‘nutrient’ ‘vitamin’ ‘micronutrient’ and ‘bioactive’ is inconsistent throughout the text.  A vitamin is a nutrient (defined as compounds in foods essential to life and health), specifically it is a micronutrient (a nutrient that is needed in small amounts daily).  EGCG and Res are not considered ‘nutrients’ but are active in metabolism (hence termed bioactives).  The authors are advised to use these terms correctly and consistently throughout the text. For example the statement at line 1 page 17 is awkward since vitamins are nutrients:  In this study we have evidenced the interactions between nutrients and vitamins.

Please clarify the statement poor cellular ‘loading’ at line 200 and provide a citation.  Why is this not the case for the other agents as well? 

Please provide more detailed information on the concentration of nutrients and bioactives used in these stimulation assays – and compare to physiological concentrations. Line 19 page 17

Which stereoisomer of alpha tocopherol was used in these assays?    Line 18 page 17

The discussion should contain a limitations section to make the reader aware of the technical limitations inherent in these in vitro assays.   

Author Response

Reviewer 1

The impact of oral nutrients and bioactives on immune responses is an important focus deserving of much coverage in the research literature.  This paper contributes to this topic.  A few items that should be addressed by the authors are listed below. 

The use of the terms ‘nutrient’ ‘vitamin’ ‘micronutrient’ and ‘bioactive’ is inconsistent throughout the text.  A vitamin is a nutrient (defined as compounds in foods essential to life and health), specifically it is a micronutrient (a nutrient that is needed in small amounts daily).  EGCG and Res are not considered ‘nutrients’ but are active in metabolism (hence termed bioactives).  The authors are advised to use these terms correctly and consistently throughout the text.

Response 1: We fully agree with this and have revised the relevant terms in this form of the manuscript.

For example the statement at line 1 page 17 is awkward since vitamins are nutrients:  In this study we have evidenced the interactions between nutrients and vitamins.

Response 2: This has been done.

Please clarify the statement poor cellular ‘loading’ at line 200 and provide a citation.  Why is this not the case for the other agents as well? 

Response 3: The concept of cellular loading is based on our previous studies, which is cited as reference. Sensu stricto and to the best of our knowledge experimental data on cellular loading of bioactives have not been published. Data with ascorbic acid do underscore the concept, although an active transport mechanism is involved in the case of this vitamin.

Please provide more detailed information on the concentration of nutrients and bioactives used in these stimulation assays – and compare to physiological concentrations Line 19 page 17 .

Response 4: The concentrations used in these experiments are mentioned. The comparison with physiological concentrations is highlighted in depth in the discussion section.

Which stereoisomer of alpha tocopherol was used in these assays?    Line 18 page 17

Response 5: We used the all-rac a-tocopherol

The discussion should contain a limitations section to make the reader aware of the technical limitations inherent in these in vitro assays.

Response 6: We feel that this is widely done in the Discussion section, where the intrinsic consequences on the use of bioactives in short-term versus long-term modulation of the immune response is discussed.   

Reviewer 2 Report

This work evaluated the effects of vitamin A, D, E and Res, and EGCG on the adaptive immune response, synergistic or antagonistic interactions. Among other results, the MS show that Res and EGCG altered CD4 and CD8+ expressions, and vitamins also changed secretion and expression of interleukins, and cytokines. Th1 and Th2 response was also changed. It is concluded that Res and EGCG might interact with vitamins and support the adaptive immune response.

The work is completely unformatted in accordance with (the minimum) required by the journal. Even several different fonts are used. The MS demonstrates a lot of carelessness on the part of the authors. Besides, this work is not entirely novel.  Res and EGCG has been extensively studied.

Other points are:

Introduction:

  • This section starts abruptly and directly with the description of Th1 and Th2, followed by a description of the immune response. There is not a general sentence about immunity of similar... It does not seem OK in this reviewer’s opinion;
  • Lines 44-54: seems more like the start of the introduction;
  • Line 62: the introduction should not have any results. Remove this;

Results:

  • “Results not shown” – these results are interesting and should be in the supplementary files;
  • “2. In vitro differentiation of activated T lymphocytes” – if the title is in italic form, “in vitro” should not be (check the entire MS for this detail and correct);
  • “Cytofluorometric analysis was performed after 4 days of culture” – why particularly 4 days? Please clarify this (with reference(s));
  • Figure 1 is distorted, which difficult its analysis. Besides, it is poorly cut;
  • The descriptions on the figures should be in the figure captions, not separated;
  • Figure 2 and 4 are too small to be completely understood;
  • Significances on graphs (P value)?

M&M:

  • For every reagent and material, there is the need to be indicated country and manufacturer. Check and correct the entire MS;

Conclusion:

  • This section is too small, poor (one single sentence)... Cut or improve;

Author Response

Reviewer 2

This work evaluated the effects of vitamin A, D, E and Res, and EGCG on the adaptive immune response, synergistic or antagonistic interactions. Among other results, the MS show that Res and EGCG altered CD4 and CD8+ expressions, and vitamins also changed secretion and expression of interleukins, and cytokines. Th1 and Th2 response was also changed. It is concluded that Res and EGCG might interact with vitamins and support the adaptive immune response.

The work is completely unformatted in accordance with (the minimum) required by the journal. Even several different fonts are used. The MS demonstrates a lot of carelessness on the part of the authors.

Response 1: We disagree with this statement. We did not publish a paper but submitted it for peer review mainly. according to the ‘Instructions for authors’. The pdf for peer review is generated by MDPI software and thus not accessible to the authors.

Besides, this work is not entirely novel.  Res and EGCG has been extensively studied.

Response 2: In fact, there is a noticeable bulk of published evidence of the effect of bioactives on some parameters of the immune response. Yet, the novelty of the paper relies on the description of their interaction with vitamins.

Other points are:

 Introduction:

  • This section starts abruptly and directly with the description of Th1 and Th2, followed by a description of the immune response. There is not a general sentence about immunity of similar... It does not seem OK in this reviewer’s opinion;
  • Response 3: It has been changed.
  • Lines 44-54: seems more like the start of the introduction;
  • Response 4: It has been changed.
  • Line 62: the introduction should not have any results. Remove this;
  • Response 5: We have done this, although we think that one could argue about it.

 Results:

  • “Results not shown” – these results are interesting and should be in the supplementary files;
  • Response 6: Some parameters were used as quality check for the experimental approaches. Since they were not modulated during in vitro culture and not influenced by the substances, we did not include those (neutral) results in the paper.
  • “2. In vitro differentiation of activated T lymphocytes” – if the title is in italic form, “in vitro” should not be (check the entire MS for this detail and correct);
  • Response 7: It has been done.
  • “Cytofluorometric analysis was performed after 4 days of culture” – why particularly 4 days? Please clarify this (with reference(s));
  • Response 8: This has been corrected (to 5 days).
  • Figure 1 is distorted, which difficult its analysis. Besides, it is poorly cut; Response 9: If published this should be done by the editorial office.
  • The descriptions on the figures should be in the figure captions, not separated;
  • Figure 2 and 4 are too small to be completely understood;
  • Response 10: Final size is at the discretion of the editorial office.
  • Significances on graphs (P value)?
  • Response 11: This has been added where appropriate.

 M&M:

  • For every reagent and material, there is the need to be indicated country and manufacturer. Check and correct the entire MS;
  • Response 12: This has been done.

 Conclusion:

  • This section is too small, poor (one single sentence)... Cut or improve; Response 13: We have modified and amended this short section.

Reviewer 3 Report

The authors of the study analyzed the effect of vitamins and micor-nutrients on the adaptive immune response.

The study showed some weekness and the manuscript is not easy to read.

Major points:

1. The authors should describe the results more in detail, not only for exapample:“Res and VD mutually influenced cytokine production..“ Abstract line 27 or results line 125: „VE impacted cytokines. What means influence or impacted? It should be clear for the reader how they influence it, increase or decrease... The authors should also give numbers.How big is the difference And all results should be explained, not only some given for example like line 158.

2. Figures: The authors should add A., B, C. to the single graphes in figures. Without this it is not easy to find quickly the right graph that they describe in the text.

3. Figure legends: The authors should rewrite all figure legends. The informations are not sufficient to understand the figures without the text. Describe also the abbreviations used in the figure. Are the bars in figure 2 and 3 means?

4. In legend of figure 2, it is written that triplicate cultures were analysed? Where this experiments not done in 3 independent experiments or at least 3 different buffy coats? Triplicats are not sufficient.

5. Some parts of introduction and discussion is more superficial like the sentence in line 201. Give more details

6. Do the authors used positive controls or reference substances?

Minor points

1. Not for all materials, the company together with city and country are given. Please indicate. Use the complete company name.

2. The author should add the purity of all reagents used.

3.Explain the abbreviation EGCG in the abstract (line21)

4. line 44: The author wrote that they used „usually 5x10^5 cells. Means that they sometimes use other cell numbers? Please indicate.

5.line 58 Itis written that PGE2 is meassured. Where can I find the results for PGE2?

6.List of abreviation is not complete.

Author Response

Reviewer 3

The authors of the study analyzed the effect of vitamins and micor-nutrients on the adaptive immune response.

The study showed some weekness and the manuscript is not easy to read.

Major points:

  1. The authors should describe the results more in detail, not only for exapample:“Res and VD mutually influenced cytokine production..“ Abstract line 27 or results line 125: „VE impacted cytokines. What means influence or impacted? It should be clear for the reader how they influence it, increase or decrease... The authors should also give numbers.How big is the difference And all results should be explained, not only some given for example like line 158.
  2. Response 1: We have changed it appropriately. The effects are small in some parameters (see details in presented data) we think that their precise quantification need to be evaluated in further experiments where the culture periods and concentrations of the substances are further evaluated
  3. Figures: The authors should add A., B, C. to the single graphes in figures. Without this it is not easy to find quickly the right graph that they describe in the text.
  4. Response 2: Each panel of the Figures contains the name of the analyzed parameter.
  5. Figure legends: The authors should rewrite all figure legends. The informations are not sufficient to understand the figures without the text. Describe also the abbreviations used in the figure. Are the bars in figure 2 and 3 means?
  6. Response 3: This has been done, where appropriate
  7. In legend of figure 2, it is written that triplicate cultures were analysed? Where this experiments not done in 3 independent experiments or at least 3 different buffy coats? Triplicats are not sufficient.
  8. Response 4: The absolute values vary between individual buffy coats. We show the results of representative experimental series where triplicates or quadruplicates are standard.
  9. Some parts of introduction and discussion is more superficial like the sentence in line 201. Give more details
  10. Response 5: With regard to vitamin D (line 201), there is a plethora of published evidence that we felt not necessary to review in more details
  11. Do the authors used positive controls or reference substances?
  12. Response 6: Yes in all cases where appropriate

Minor points

  1. Not for all materials, the company together with city and country are given. Please indicate. Use the complete company name.
  2. Response 7: This has been done.
  3. The author should add the purity of all reagents used.
  4. Response 8: This has been done.

3.Explain the abbreviation EGCG in the abstract (line21):

Response 9: This has been done.

  1. line 44: The author wrote that they used „usually 5x10^5 cells. Means that they sometimes use other cell numbers? Please indicate.
  2. Response 10: The same numbers of cells have been used.

5.line 58 Itis written that PGE2 is meassured. Where can I find the results for PGE2?

Response 11: This has been removed.

6.List of abreviation is not complete.

Response 12: This has been corrected.

Round 2

Reviewer 2 Report

1.     Use “epigallocatechin-3-gallate”, not “epigallo catechin-3-gallate”. When the abbreviation form is indicated (abstract), the authors do not need to indicate again (introduction);

2.     “Cytofluorometric analysis was performed after 4 days of culture” – why particularly 4 days? Please clarify this (with reference(s));

Response 8: This has been corrected (to 5 days).

This still does not clarify the choice. Rational for 4-5 days? Reference(s) to support this? This needs to be clearly indicated.

3.     Figure 3: SD and p values (significance) need to be indicated and discussed.

4.     Statistical software used (brand, country and version)?

5.     Conclusion could bring more particular and general closing insights, but this is just an opinion.

6.     Final comment: the tone of the authors' replies seems to show that they do not recognize that their MS has failures. This reviewer reminds that we all have less right points in our MS (and this reviewer reinforces: all of us), and the debate of those details in a peer review process should be done with politeness, which was not the case, and it is quite unfortunate.

Author Response

Reviewer 2

  1.     Use “epigallocatechin-3-gallate”, not “epigallo catechin-3-gallate”. When the abbreviation form is indicated (abstract), the authors do not need to indicate again (introduction);

This has been done as required.

  1.     “Cytofluorometric analysis was performed after 4 days of culture” – why particularly 4 days? Please clarify this (with reference(s)); This still does not clarify the choice. Rational for 4-5 days? Reference(s) to support this? This needs to be clearly indicated.

This had been corrected to 5 days. The culture period of 5 days for phenotypic analysis of differentiated T lymphocytes is consistent with published data (where 4-6 days were frequently used in experimental protocols; see also Xin-Hua Feng et al (eds.), TGF-b Signaling Methods and Protocols Methods in Molecular Biology, vol 1344, DOI 10.10007/978-1-4939-2966-2, Springer Science-Business Media New York, 2016). We have added reference 20 (i.e. Falchetti et al. 2001) at appropriate places of the manuscript.

  1.     Figure 3: SD and p values (significance) need to be indicated and discussed.

This has been done as required.

  1.     Statistical software used (brand, country and version)?

This has been done as required.

  1.     Conclusion could bring more particular and general closing insights, but this is just an opinion.

We would ask the editor of the special issue whether another version of ‘Conclusions’ is required.

  1.     Final comment: the tone of the authors' replies seems to show that they do not recognize that their MS has failures. This reviewer reminds that we all have less right points in our MS (and this reviewer reinforces: all of us), and the debate of those details in a peer review process should be done with politeness, which was not the case, and it is quite unfortunate.

We do fully agree with the reviewer and recognize that the MS had (and still might have) failures. We regret that the author of the previous reply (JS) did not show enough respect for the very valid work done by the reviewer on a voluntary basis.

Reviewer 3 Report

The authors improved the manuscript, but few minor points were not implement.

1.The authors should add A., B, C. to the single graphes in figures. Without this it is not easy to find quickly the right graph that they describe in the text.

2. Figure legend. All used abbreviations in the figure should be explained.

3. The answer to the question 2Do the authors used positive controls or reference substances?" was yes, but do the authors include the conntrols in the graphes?

4. Why id the authors use in one case SEM and another SD?

In the next response, the author should give page or line number where they changed the cprresponding item.

Author Response

Reviewer 3

The authors improved the manuscript, but few minor points were not implement.

1.The authors should add A., B, C. to the single graphes in figures. Without this it is not easy to find quickly the right graph that they describe in the text.

This has been done and the modified figures were also sent to the editor in separate files

  1. Figure legend. All used abbreviations in the figure should be explained.

We feel that this contrasts with the requirements of reviewer 2. Therefore, we ask the editor to choose the appropriate version.

  1. The answer to the question 2 Do the authors used positive controls or reference substances?" was yes, but do the authors include the conntrols in the graphes?

This has now been described more extensively.

  1. Why id the authors use in one case SEM and another SD?

SEM (standard error of the means) was calculated when the results obtained from several independent experiments (i.e. PBMCs from different donors), done in triplicates, were used. SD (standard deviation) refers to the value obtained from triplicate cultures done with PBMCs from a given donor.

In the next response, the author should give page or line number where they changed the cprresponding item.

The Markups option used in the revised MS should permit to follow the changes in the text. Changes made during Revision II are indicated in blue